# Evaluation of lay health workers on quality of care in the inpatient setting

Ramona Basnight[1◉], Peter Berry[2◉], Kellie Capes[3◉], Sherri Pearce[4◉], Julie Thompson[5◉], Deborah H. Allen[6◉], Bradi B. Granger[5◉], Staci S. Reynolds[5◉] *

1 Duke Cancer Institute, Duke Raleigh Hospital, Raleigh, North Carolina, United States of America, 2 General Medicine, Critical Care, and Emergency Services, Duke Raleigh Hospital, Raleigh, North Carolina, United States of America, 3 Medical Surgical Department, Duke Regional Hospital, Durham, North Carolina, United States of America, 4 Musculoskeletal/Spine Department, Duke University Health System, Durham, North Carolina, United States of America, 5 Duke University School of Nursing, Durham, North Carolina, United States of America, 6 Nursing Research and Evidence-Based Practice, Duke University Health System, Durham, North Carolina, United States of America

◉ These authors contributed equally to this work.
* staci.reynolds@duke.edu

## Abstract

### Aims

To evaluate the impact of a lay health worker support role in the inpatient setting.

### Background

Healthcare systems are facing critical nursing and nurse assistant staffing shortages. These disciplines can be challenging to recruit and retain, leading healthcare leaders to identify innovative staffing models. Whereas lay health workers have been used in the community and low-income setting, there is scant evidence of their use in the inpatient setting. We implemented a lay health worker role, called Patient Attendant Service Aides (PASAs), on two medical/surgical units at a community hospital.

### Methods

A pre/post-implementation design was used for this study. An online survey was provided to nurses, nursing assistants, and PASAs on the two medical/surgical units to assess their satisfaction and perceptions of the role. Nursing quality metrics, patient satisfaction, and nursing and nursing assistant turnover were evaluated before and after implementing the role.

### Results

The online survey showed that nurses and nursing assistants felt that PASAs helped offload their workload, allowing them to focus on nursing-related tasks. PASAs felt supported by the team and believed they were making a meaningful contribution to the unit. There were slight improvements in patient satisfaction, although not significant. There was a significant improvement in nursing turnover on Unit A, from 71.1% to 21.6% (p = 0.009).

**Data Availability Statement:** All relevant data are within the paper and its Supporting Information files.

**Funding:** The author(s) received no specific funding for this work.

**Competing interests:** The authors have declared that no competing interests exist.

## Conclusions

This is one of the first studies to evaluate the use of lay health workers in the inpatient setting; we found this role to be a feasible way to offload tasks from clinical staff. This role may serve as a pathway for workforce development, as several PASAs are now enrolled in nursing assistant training. Nurse managers may consider using lay health workers in the inpatient setting as they face severe clinical staff shortages.

## Introduction

Currently, nursing shortages are a major concern among healthcare facilities. Although cyclical nursing shortages have occurred in the past, the COVID-19 pandemic, combined with the baby boomer generation becoming a larger sector of the healthcare consumer market, has exacerbated the acute nursing shortage in the past two years [1]. The inability to provide adequate nursing resources has been shown to have an impact on quality measures such as healthcare-associated infections (HAIs), patient satisfaction, mortality, and safety related patient outcomes [2–4]. Studies by Ball et al. [5] and Griffiths et al. [6] confirmed that many nurses report omissions of care when staffing levels are inadequate. In their review of current research, Griffiths et al. [6] noted that 14 studies found low nurse staffing levels were significantly associated with higher reports of missed care. Staffing shortages can also have an impact on nurse satisfaction, morale, and burnout, which in turn has led to more nurses leaving the bedside [7, 8]. As experienced nurses continue to leave bedside nursing, the inability to meet the demand for care has resulted in a cycle of shortage-induced resignations and retirements [9].

To counteract these shortages, many institutions have turned to use of agency or travel nurses to fill the gap, resulting in financial hardships and in some cases, hospital closures. The costs of using agency/travel nurses and staffing incentives are not sustainable, especially when combined with cancellation of elective surgeries to create capacity, costs of providing care and personal protective equipment, and overall reductions in healthcare reimbursement during the COVID-19 crisis all resulting in lost sources of revenue for many healthcare organizations [10, 11]. As such, alternate staffing models need to be identified by healthcare systems.

During COVID-19, crisis staffing models were implemented to support patient care by maximizing the available nurse (i.e., making them more capable) and nursing support staff. Team nursing, buddy systems, and increased use of unlicensed assistive personnel are a few examples of these strategies, many of which were successful in meeting the acute needs of those patients. However, those models also served to increase burnout and stress among the staff and were abandoned as COVID-19 hospitalizations decreased [12–14]. Although COVID-19 related admissions are decreasing, the nursing shortage is expected to remain and potentially increase in the next decade.

Several studies have suggested alternate staffing models through increased use of nursing assistants [15, 16]. However, hospitals have also experienced a significant decrease in nursing assistant staff during the current nurse staffing shortage. Efforts to develop alternate care models or add nursing assistants to offset nursing vacancies have been impacted by the lack of available staff. Shang et al. [4] recommended further study of the impact of low levels of certified nursing support staffing on patient safety, as much of the work has shifted from nursing assistants to nursing staff. This in turn reduces the nurse's ability to complete other work, such as assessment and monitoring [4].

Previous community researchers have utilized lay health workers to help fill clinical staffing shortages [17–19]. Lay health workers are defined as "any health worker carrying out functions related to healthcare delivery; trained in some way in the context of the intervention; and having no formal professional or paraprofessional certificated and degreed tertiary education" [18]. Lay health workers can offload work previously completed by clinical staff, allowing staff the opportunity to work at the top of their license [17]. Whereas lay health workers have been used in community settings as care guides, there is no evidence on use of lay health workers in the hospital inpatient setting. Innovative models that explore the use of non-traditional, non-clinical roles are needed to support patient care and improve the work environment for current and future nurses.

Due to challenges with registered nurse (RN) retention and nursing assistant recruitment at a community hospital in the Southeastern United states, leadership, nursing recruitment, and workforce development stakeholders identified that hiring lay health workers with no medical background had a high yield for recruitment. As such, an innovative Patient Attendant Safety Aides or PASA role for the inpatient setting was developed. For this role, individuals did not need to have any clinical experience or healthcare background. They were recruited through online career sites and the healthcare system's recruitment website, with pay comparable to nursing assistants. Individuals were selected for the PASA role if they had a high school diploma or equivalent; prior health care experience was not required, but an interest in working with people in a health care setting was needed. Applicants hired into this role were scheduled to work eight-hour shifts.

To determine the tasks that PASAs would complete, a list of tasks appropriate for lay health workers was provided to the nursing staff on two medical/surgical units that piloted the role. Each nurse had an opportunity to identify the top three tasks they felt would be most impactful for PASAs to complete. Final identified tasks included assisting with ambulation, turning, toileting, and feeding patients. Based on this task list, a comprehensive program was developed in collaboration with the hospital's clinical education department to train the PASAs in these tasks. Training was completed by having PASAs attend hospital-based orientation, followed by a 4-week competency-based orientation that was approved by the clinical education department. Preceptors who were assigned to work with the PASAs one-to-one completed competency assessments using a teach-back method by observing return demonstrations of tasks. During these 4 weeks, PASAs also spent time shadowing nursing assistants to develop an understanding of operational flow on the unit. PASAs were provided with additional training for specific tasks (e.g., ambulation, turning) by subject matter experts in physical therapy and wound ostomy care.

## Purpose

The purpose of this study was to evaluate the impact of the PASA role on quality of care. The specific aims were to: (1) evaluate staff satisfaction and perception of the PASA role; (2) evaluate the impact of the PASA role on nursing quality metrics; (3) evaluate patient satisfaction scores; and (4) evaluate RN and nursing assistant turnover.

## Methods

### Design

A pre/post-implementation design was used for this study to evaluate the impact of the PASA role on quality of care. The PASA role began in January 2022. Pre-implementation data were collected from January to December 2021, with post-implementation data collected from

January to July 2022. This study was determined exempt by the university's Institutional Review Board and deemed to not be human subjects' research.

## Setting and sample

This study was conducted within two medical/surgical units at a community hospital in the Southeastern United States. The hospital is part of a large academic health system and includes 186 beds; the medical/surgical units had a total of 64 beds, with 98 RNs and 62 nursing assistants. A total of ten PASAs began working on the two medical/surgical units in January 2022 (five PASAs on each unit).

## Variables

To evaluate the impact of the PASA role, we assessed staff (RN, nursing assistant, PASA) satisfaction and perceptions of the PASA role, nursing quality metrics, patient satisfaction scores, and RN/nursing assistant turnover rates. All data were provided in de-identified, aggregate form. Staff satisfaction and perceptions of the PASA role were measured via an anonymous, electronic survey provided to all RNs, nursing assistants, and PASAs on the two pilot units. The survey included 11 questions for RNs and nursing assistants, with nine questions for the PASAs. Staff were asked to rate questions on a 4-point Likert scale (1 = strongly disagree to 4 = strongly agree). The survey was developed by content experts on the research team, including four Doctor of Nursing Practice-prepared health system operational leaders and three PhD-prepared nurse scientists. Content validity of each item was ascertained by expert review. No additional testing of the survey was conducted prior to distribution. The survey was disseminated to respondents via Quick Response (QR) codes; staff were asked to complete the survey between October 17 to 31, 2022.

Five nursing quality metrics were assessed before and after implementation of the PASA role, including central line-associated bloodstream infections (CLABSI), catheter-associated urinary tract infection (CAUTI), *Clostridioides difficile* (*C. diff*) infections, patient falls, and healthcare-associated pressure injuries. Healthcare-associated infections (CLABSI, CAUTI, *C. diff*) were measured per National Healthcare Safety Network (NHSN) standard definitions [20]. Rates of infections (per 1000 device or patient days) were provided by the hospital's infection prevention department on a monthly basis. Whereas PASAs did not have a direct role in infection prevention measures, healthcare-associated infections were included in data analysis to understand if the PASA role off-loaded tasks from the RN, allowing the RN more time to focus on infection prevention measures. Patient falls were defined as an unplanned patient descent to the floor [21]; these data were collected by the hospital's internal safety reporting system and reported as a rate of falls per 1000 patient days. Healthcare associated pressure injuries included in this study were all pressure injuries over a boney prominence. These data were collected by the nurses and/or wound ostomy nurses and put into an internal safety reporting system; these injuries were measured as a rate per 1000 patient days.

Three questions from the Hospital Consumer Assessment of Healthcare Providers and Systems (HCAHPS) survey were used to measure patient satisfaction, including: "How well staff worked together to care for you" (Likert-scale of 1 = very poor to 5 = very good); "During this hospital stay, after you pressed the call button, how often did you get help as soon as you wanted it?" and "How often did you get help in getting to the bathroom or in using a bedpan as soon as you wanted?" (Likert-scale of 1 = never to 4 = always). A score for the last two questions was aggregated by HCAHPS and provided data on the overall "Responsiveness of Hospital Staff." These data were collected by anonymous patient satisfaction surveys sent by a third-party vendor to patients. Data were available in aggregate form from the health system's

internal data warehouse and reported as the number of responses with a rating of "Always" over the total responses. Nursing and nursing assistant turnover were defined as the percent of RNs and nursing assistants who left the health system. These data were collected by the hospital's Human Resources Department.

## Data analysis

SPSS Version 28 was used for all data analysis, with an alpha of .05 determined to be significant. Descriptive statistics (frequency, mean, standard deviation [SD]) were used to evaluate staff satisfaction and perception of the PASA role. Quality metrics, patient satisfaction scores, and RN/nursing assistant turnover were analyzed using Mann-Whitney U tests.

## Results

Staff satisfaction and perception survey results are reported in Table 1. A total of 99 staff completed the survey, 57.6% (n = 57) RNs, 35.4% (n = 35) nursing assistants, and 7.1% (n = 7) PASAs, for a 58.2% response rate.

Overall, the RNs and nursing assistants expressed satisfaction with the PASA role (mean = 3.06, SD = 0.81). Although many RNs and nursing assistants agreed that the PASA role decreased their workload (mean = 2.90, SD = 0.75), they rated adequacy of staffing lower (mean = 2.65, SD = 0.73). Many RNs and nursing assistants (n = 33) felt that the PASA role could take on additional tasks, such as checking fall alarms, replenishing supplies, documenting intake and output, and cleaning up incontinence episodes. Due to the PASA role offsetting tasks, many RNs and nursing assistants felt that they could better use their skills (mean = 3.03, SD = 0.81). Open-ended comments revealed that, whereas the PASA role provided beneficial

**Table 1. Staff satisfaction and perceptions of the PASA role.**

| Question | N (%), Mean (SD) |
|---|---|
| **Role** | |
| Registered Nurse (RN) | 57 (57.6%) |
| Nursing Assistant (NA) | 35 (35.4%) |
| PASA | 7 (7.1%) |
| **Questions for RNs and NAs only** | |
| Overall satisfaction with the PASA role | 3.06 (0.81) |
| The PASA role has decreased my workload | 2.90 (0.75) |
| My work unit is adequately staffed | 2.65 (0.73) |
| Having the support of the PASA role has impacted my decision to remain employed on this unit | 2.59 (0.82) |
| I feel the PASAs tasks (toileting, turning feeding, and ambulating) appropriately meet the demands of the unit | 2.91 (0.78) |
| I can better use my skills due to the PASA role offsetting other tasks | 3.03 (0.81) |
| I have sufficient time to provide the best care/service for our clients/patients due to the PASA role offsetting other tasks | 2.83 (0.74) |
| **Questions for PASAs only** | |
| Overall satisfaction with the PASA role | 3.86 (0.35) |
| I received the training I needed to do a good job in the PASA role | 3.57 (0.49) |
| I have a sense of belonging on the unit in the PASA role | 3.43 (0.73) |
| I feel supported by my team members (RNs and NAs) | 3.57 (0.49) |
| I have adequate time to complete the daily tasks I am assigned | 3.71 (0.45) |
| This role provides me with future career opportunities in this organization | 3.43 (0.73) |

**Table 2. Nursing quality, patient satisfaction, and turnover findings before and after implementation of the PASA role.**

| Variable | Unit A | | | Unit B | | |
|---|---|---|---|---|---|---|
| | **Pre** | **Post** | **p-value** | **Pre** | **Post** | **p-value** |
| **Quality metric** | | | | | | |
| Fall rate median | 1.86 | 3.06 | 0.076 | 2.34 | 2.36 | 0.472 |
| Pressure Injury rate median | 0.47 | 0.0 | 0.452 | 0.4 | 0.1 | 0.861 |
| **Patient satisfaction** | | | | | | |
| Work together | 62.8% | 70% | 0.063 | 73.9% | 75% | 0.671 |
| Help toileting | 60% | 55.6% | 0.233 | 63.4% | 64% | 0.866 |
| Call light responsiveness | 50% | 52.2% | 0.394 | 58.7% | 64.7% | 0.220 |
| Responsiveness | 51.8% | 54.2% | 0.310 | 64.8% | 69.5% | 0.398 |
| **Turnover** | | | | | | |
| Registered nurse | 71.1% | 21.6% | **0.009** | 32.2% | 35.3% | 0.398 |
| Nursing assistant | 65.7% | 74.2% | 0.673 | 65.3% | 70.6% | 0.398 |

support on shifts, there are areas of opportunity for additional training of PASAs to have a better understanding of the healthcare environment and patient needs.

From the PASA's perspective, many were very satisfied with the role (mean = 3.86, SD = 0.35). They felt they received the training needed to do a good job (mean = 3.57, SD = 0.49) and felt supported by their team members (mean = 3.57, SD = 0.49). Open-ended comments revealed that they enjoy the role and feel that they are making a meaningful contribution to the workforce. One recommendation was to allow PASAs to work 12 hour shifts that mirror RN and nursing assistant shifts, rather than the current eight-hour shifts.

As HAIs are a rare event for both units, CAUTIs, CLABSIs, and *C. diff* rates were not statistically analyzed due to lack of variation (S1 Fig). Rates of HAIs remained low or nonexistent before and after implementation of the PASA role on both units. For Unit A, a Mann-Whitney U test showed the median fall rate increased from pre (1.86/1000 patient days) to post-implementation of the PASA role (3.06 falls/1000 patient days), but it was not statistically significant (p = .076). For Unit B, the median fall rate increased slightly from pre (2.34/1000 patient days) to post-implementation (2.36/1000 patient days), but was also not statistically significant (p = .472). For healthcare associated pressure injuries, Unit A's median pressure injury rate declined from pre (.47/1000 patient days) to post-implementation (0.0 /1000 patient days), but it was not statistically significant (p = .452). For Unit B, the median pressure injury rate increased from pre (.4/1000 patient days) to post-implementation (.71 /1000 patient days), but was also not statistically significant (p = .861) (Table 2).

Patient satisfaction ratings slightly increased for most questions in both Units A and B; however, these differences were not significant (Table 2, Fig 1 Patient satisfaction scores). RN turnover significantly decreased in Unit A from 71.1% to 21.6%, p = 0.009. Nursing assistant turnover on Unit A, as well as RN and nursing assistant turnover on Unit B, increased slightly post-implementation; however, this was not significant (Table 2).

## Discussion

This study found a significant reduction in RN turnover on Unit A, and overall staff satisfaction and perceptions of the PASA role were positive. However, we did not find any significant improvements in quality metrics, and only slight improvements in patient satisfaction scores after implementing the PASA role. To the best of our knowledge, this is one of the first studies that has sought to evaluate the impact of lay health workers in the inpatient setting as an

Unit A

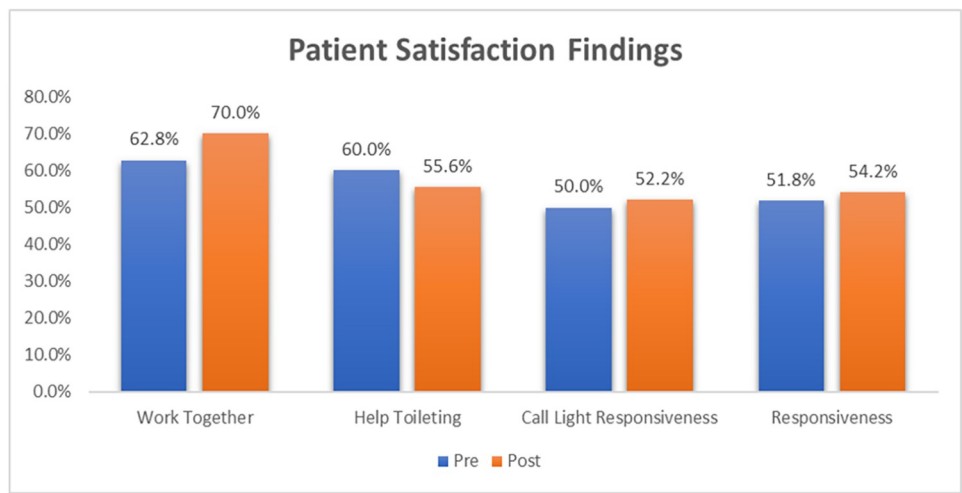

Unit B

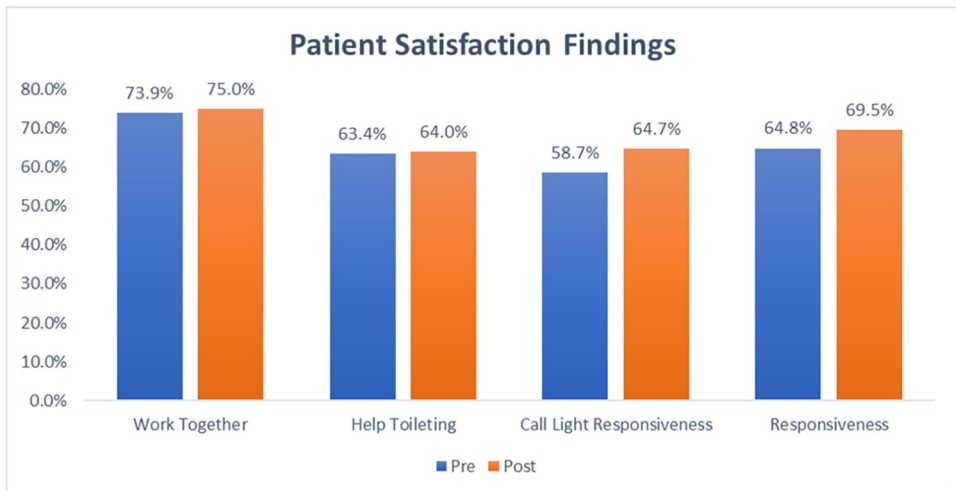

**Fig 1. Patient satisfaction scores.**

alternative staffing model; as such, these findings contribute to our knowledge regarding innovative staffing models.

Previous research has illustrated the use of lay health workers in community, rural, and low-income settings, often as care guides to assist with coordination of patient care with healthcare providers or provide education and social support [17, 19, 22]. Cain and colleagues [17] conducted semi-structured interviews to examine the use and experiences of lay health workers that served as support for patients with serious illnesses. They found that lay health workers did not feel adequately supported by team members, which created a potentially toxic environment [17]. This finding is different from ours, as PASAs positively responded that they felt supported by their team members. This may be due to the administrative and nurse manager support of the PASA role, which allowed nursing staff to identify the top tasks that the PASAs should complete, providing a sense of guidance and support of the role.

Patel and colleagues [23] similarly assessed patient satisfaction six months after implementing lay health workers in the outpatient setting to assist with documentation of end-of-life care preferences. They found a statistically significant improvement in patient satisfaction with the item "satisfaction with provider" (p<0.001) and an increase in documentation of goals of care (p<0.001). While we did not observe a statistically significant improvement in patient satisfaction scores after the PASA role was implemented, there was a slight improvement in the majority of HCAHPS scores. Evaluating patient satisfaction over a longer period of time may demonstrate role success and significant outcome improvements. We also found a significant reduction in RN turnover on Unit A; however, this finding must be interpreted with caution. These data were obtained from the Human Resources Department and the reasons for resignations were many and varied. The PASA role may have had limited impact on RN's rationale for leaving the unit. Further research is needed to understand the impact of this type of role on turnover rates.

We observed an increase in fall rates on both units after implementing the PASA role. This could be due to the fact that ambulation was occurring more frequently, as ambulation was a task assigned to the PASA. Additionally, as tasks were off-loaded from the RN and nursing assistants, they too may have had more time to ambulate their patients. Patey and colleagues [24] also saw an increase in falls after implementing a mobility program. However, many factors influence fall rates. For example, Mikos and colleagues [25] found that most patient falls in the inpatient setting occur between midnight and six in the morning; many PASAs did not work during this timeframe. As such, these findings may be interpreted positively but with reservation.

Lastly, Karungi and colleagues [22] conducted a pre/post-intervention study to examine lay health worker's knowledge of dementia care before and after an educational intervention. After education, they found that lay health workers had a more basic understanding of dementia and felt more comfortable caring for this patient population [22]. We did not conduct a pre/post-analysis of PASA knowledge after they were provided training on the role; however, future studies may consider measuring knowledge and comfort as an outcome.

## Limitations

This study has several limitations. Only five PASAs were employed on each unit, each of whom worked eight-hour shifts; as such, there was not 24/7 coverage of shifts with this role. Hiring more PASAs to allow coverage during all shifts may have more of an impact on quality metrics. Additionally, the role was evaluated for only seven months after implementation; a longer timeframe for evaluation may yield different results. The survey that evaluated staff satisfaction and perceptions of the PASA role was developed by the research team and was only tested among team member experts for content validity. The survey did not undergo further psychometric testing; future studies may consider using a valid and reliable survey. Lastly, this study was conducted at only one community hospital within two medical/surgical units, which limits generalizability. Future research should be conducted using the role in a larger setting.

## Conclusions

This study represents one of the first to evaluate the use of lay health workers in the inpatient setting as an innovative staffing model to help impact the nursing shortage. We found using lay health workers in the inpatient setting to be a feasible way to offload tasks from clinical staff, allowing them to focus on nursing-related skills. Improvements in quality metrics were not realized in the short post-implementation period; however, patient satisfaction improved.

Additionally, RNs, nursing assistants, and PASAs were satisfied with the innovative role. Future research should continue to evaluate this role to assess for effectiveness and may consider measuring lay health worker's knowledge and comfort in the clinical setting, as well as missed nursing care and/or compliance with evidence-based practices.

## Implications for nursing management

This project offers an alternate strategy for nurse leaders to address the current staffing crises. Based on the process analysis and initial learnings, we are able to show this role could have a multitude of positive effects on nursing unit staffing and the staff's experience in the work environment. As it relates to staff retention, staff responses indicate that job satisfaction related to adequacy of staffing, teamwork, and the ability to feel that they have provided appropriate care to their patients are improved. As nurse leaders continue to support and empower all staff to work to the top of their scope, this project supports the right person doing the right job. The PASA role mitigates the transfer of nursing assistant tasks to the nurses and prevents the need for RN staff to "prioritize" technical tasks versus providing nurse licensed care such as patient teaching or care coordination.

This role also supports the evaluation of an innovative solution that has not been widely disseminated. Secondary to the staff shortage, other models that include licensed practical nurses and other unlicensed assistive personnel have been difficult to resource. This model taps into a previously underutilized opportunity to invite lay health workers to enter the healthcare career portal and could lead to further workforce development or career growth into the nursing pathway. Indeed, three PASAs are currently enrolled in nursing assistant training due to their experience in healthcare. Lastly, but of critical importance, this additional resource can provide support to meeting our patient expectation for response to their personal needs, as well as improving the adherence to our quality and safety processes to prevent pressure injuries and patient falls.

## Supporting information

**S1 Fig. Healthcare-associated infections for Units A and B.**
(DOCX)

## Acknowledgments

The authors wish to acknowledge the following individuals for their support and assistance with implementation of the lay health worker role: Kelly Kettler, MSN, APRN, ACCNS-AG, Rhea Batten, MSN, RN, and Ronda Decker, MSN, RN, CNML.

## Author Contributions

**Conceptualization:** Ramona Basnight, Peter Berry, Kellie Capes, Sherri Pearce, Deborah H. Allen, Bradi B. Granger, Staci S. Reynolds.

**Data curation:** Ramona Basnight, Peter Berry, Kellie Capes, Julie Thompson.

**Formal analysis:** Peter Berry, Julie Thompson, Staci S. Reynolds.

**Investigation:** Staci S. Reynolds.

**Methodology:** Ramona Basnight, Peter Berry, Sherri Pearce, Julie Thompson, Deborah H. Allen, Bradi B. Granger, Staci S. Reynolds.

**Project administration:** Ramona Basnight, Peter Berry, Sherri Pearce.

**Supervision:** Peter Berry, Sherri Pearce, Deborah H. Allen, Bradi B. Granger, Staci S. Reynolds.

**Visualization:** Julie Thompson, Staci S. Reynolds.

**Writing – original draft:** Kellie Capes, Staci S. Reynolds.

**Writing – review & editing:** Ramona Basnight, Peter Berry, Sherri Pearce, Julie Thompson, Deborah H. Allen, Bradi B. Granger.

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
