## [Decision Letter · Decision Letter 0]

18 Sep 2023

PONE-D-23-09485Evaluation of lay health workers on quality of care in the inpatient settingPLOS ONE

Dear Dr. Reynolds,

Thank you for submitting your manuscript to PLOS ONE. After careful consideration, we feel that it has merit but does not fully meet PLOS ONE’s publication criteria as it currently stands. Therefore, we invite you to submit a revised version of the manuscript that addresses the points raised during the review process.

We look forward to receiving your revised manuscript.

Kind regards,

Stefan Grosek, Ph.D., M.D.,

Academic Editor

PLOS ONE

Journal Requirements:

Additional Editor Comments:

Dear Authors

Your article has been reviewed by external reviewers They suggest minor revisions. Please respond to all comments point by point.

Kind regards

Reviewers' comments:

Reviewer's Responses to Questions

**Comments to the Author**

1. Is the manuscript technically sound, and do the data support the conclusions?

Reviewer #1: Yes

Reviewer #2: Partly

2. Has the statistical analysis been performed appropriately and rigorously? 

Reviewer #1: Yes

Reviewer #2: I Don't Know

3. Have the authors made all data underlying the findings in their manuscript fully available?

Reviewer #1: Yes

Reviewer #2: Yes

4. Is the manuscript presented in an intelligible fashion and written in standard English?

Reviewer #1: Yes

Reviewer #2: Yes

5. Review Comments to the Author

Reviewer #1: Dear authors many thanks for your valuable work. This study was used to evaluate the impact of a lay health worker support role in the inpatient setting. They used pre and POST design using online survey. They found that this role to be a feasible way to offload tasks from clinical staff. The topic is interesting for global reader. The manuscript was well written in a simple logical way. They consider the research ethics and they mentioned research limitations and implications.

Reviewer #2: Dear authors.

The article and the research are a welcome contribution in an era of ever-increasing labor shortages in inpatient settings. I suggest some minor corrections for the wider applicability of the content.

1 The abstract provides a summary of what was done and the key findings.

2. Lay health workers (PASA) play a key role in the research, so it would be reasonable to supplement the Introduction with a brief presentation on how they are selected to work in the inpatient setting (criteria), what are the preparations or training for performing certain tasks, e.g. Perhaps the explanation of the purpose of the research in the Introduction could be adapted (shortened) and the section in Methods from line 131 to 144 inclusive presented in the Introduction.

3. It may not be clear to the reader which of the employees of the included units participated in the electronic survey - more detailed description of the sample ( - all employed healthcare workers on the units, all present nursing staff … in adition of PASA?).

How was the survey questionnaire designed? Has the questionnaire been pre-tested?

4. It would make sense to move the line 189 in the sample description (section Methods).

By comparing Table 1 (203) and interpreting the results (line 192 -199), it is not possible to discern which questions were included in the electronic survey.

5. Perhaps some caution is needed in the interpretation of the result regarding the decrease in turnover in unit A during the time when the PASA role was included in the work of the unit; the data were obtained from the hospital's Human Resources Department and I belive that the reasons for resignations were many and varied. The conclusion based on the thus obtained data in line 233 might be supported by other sources.

Factors other than just the more frequent ambulation or mobility program can be considered in increasing the fall rate.

For example, M. Mikos et al (https://www.ncbi.nlm.nih.gov/pmc/articles/PMC8346045/) note that most patients' falls in the inpatient setting occur between midnight and 6 a.m. You also noted in the Limitations of the study that there was no coverage of lay health workers (due to limited numbers) in all work shifts.

Overall, I believe that the study opens a new field in patient care, contributes to new knowledge, and that extended research is also needed in the field of innovative employment systems in inpatient settings, which is currently a major and urgent challenge. The existing pilot contribution could be improved with additional explanations and additions, perhaps as highlighted above.

6. PLOS authors have the option to publish the peer review history of their article (what does this mean?). If published, this will include your full peer review and any attached files.

Reviewer #1: No

Reviewer #2: No

---

## [Author Response · Author response to Decision Letter 0]

2 Oct 2023

Reviewer #1: Dear authors many thanks for your valuable work. This study was used to evaluate the impact of a lay health worker support role in the inpatient setting. They used pre and POST design using online survey. They found that this role to be a feasible way to offload tasks from clinical staff. The topic is interesting for global reader. The manuscript was well written in a simple logical way. They consider the research ethics and they mentioned research limitations and implications. Thank you for your comments!

Reviewer #2: Dear authors. The article and the research are a welcome contribution in an era of ever-increasing labor shortages in inpatient settings. I suggest some minor corrections for the wider applicability of the content.

1 The abstract provides a summary of what was done and the key findings. Thank you for your comments!

2. Lay health workers (PASA) play a key role in the research, so it would be reasonable to supplement the Introduction with a brief presentation on how they are selected to work in the inpatient setting (criteria), what are the preparations or training for performing certain tasks, e.g. Perhaps the explanation of the purpose of the research in the Introduction could be adapted (shortened) and the section in Methods from line 131 to 144 inclusive presented in the Introduction. Thank you for this comment; the information from lines 131-144 have been moved to the Introduction (lines 112-136). Further information on selection criteria and training have been included in lines 119-121 (clean version)

3. It may not be clear to the reader which of the employees of the included units participated in the electronic survey - more detailed description of the sample ( - all employed healthcare workers on the units, all present nursing staff … in adition of PASA?).

How was the survey questionnaire designed? Has the questionnaire been pre-tested? Thank you for this comment; we have clarified who participated in the survey (line 160) and how it was developed (line 163-166). We have also included the fact that the survey was researcher-developed with no prior testing in the Limitations section (lines 295-298).

4. It would make sense to move the line 189 in the sample description (section Methods).Thank you; information in line 189 has been moved to the Methods section under Setting and Sample (lines 153-154)

By comparing Table 1 (203) and interpreting the results (line 192 -199), it is not possible to discern which questions were included in the electronic survey. Thank you for this comment; we have moved Table 1 up before the interpretation of the results (line 200)

5. Perhaps some caution is needed in the interpretation of the result regarding the decrease in turnover in unit A during the time when the PASA role was included in the work of the unit; the data were obtained from the hospital's Human Resources Department and I belive that the reasons for resignations were many and varied. The conclusion based on the thus obtained data in line 233 might be supported by other sources. Thank you for this thoughtful comment; we have added information in the Discussion on how these results must be interpreted with caution (lines 271-275).

Factors other than just the more frequent ambulation or mobility program can be considered in increasing the fall rate.

For example, M. Mikos et al (https://www.ncbi.nlm.nih.gov/pmc/articles/PMC8346045/) note that most patients' falls in the inpatient setting occur between midnight and 6 a.m. You also noted in the Limitations of the study that there was no coverage of lay health workers (due to limited numbers) in all work shifts. Thank you for this comment; we have added more information in the Discussion section regarding this finding (lines 280-283).

Overall, I believe that the study opens a new field in patient care, contributes to new knowledge, and that extended research is also needed in the field of innovative employment systems in inpatient settings, which is currently a major and urgent challenge. The existing pilot contribution could be improved with additional explanations and additions, perhaps as highlighted above. Thank you for this comment!

---

## [Editor Report · Decision Letter 1]

5 Oct 2023

Evaluation of lay health workers on quality of care in the inpatient setting

PONE-D-23-09485R1

Dear Dr. Reynolds,

We’re pleased to inform you that your manuscript has been judged scientifically suitable for publication and will be formally accepted for publication once it meets all outstanding technical requirements.

Kind regards,

Stefan Grosek, Ph.D., M.D.,

Academic Editor

PLOS ONE
---

## [Editor Report · Acceptance letter]

8 Oct 2023

PONE-D-23-09485R1 

Evaluation of lay health workers on quality of care in the inpatient setting 

Dear Dr. Reynolds:

I'm pleased to inform you that your manuscript has been deemed suitable for publication in PLOS ONE. Congratulations! Your manuscript is now with our production department. 

Kind regards, 

on behalf of

Professor Stefan Grosek 

Academic Editor

PLOS ONE